## [Peer Review File · Biology Open]

Gap junction-mediated signaling coordinates Rhodopsin coupling for *Drosophila* color vision

Xuanshuo Zhang, Ryoki Shinjo, Manabu Kitamata, Shinichi Otsune and Hideki Nakagoshi
DOI: 10.1242/bio.062463

Editor: Tristan Rodríguez

Review timeline

Original submission:	16 December 2025
Editorial decision:	22 December 2025
Resubmission:	4 January 2026
Editorial decision:	8 January 2026
First revision received:	10 January 2026
Accepted:	12 January 2026

Original submission

First decision letter

MS ID#: bio.062417

MS Title: Gap junction-mediated signaling coordinates Rhodopsin coupling for *Drosophila* color vision

Authors: Xuanshuo Zhang; Ryoki Shinjo; Manabu Kitamata; Shinichi Otsune; Hideki Nakagoshi

I have now reached a decision on the above manuscript.

The reviewer reports are shown at the bottom of this email.

Review of your article has raised several important concerns that, together, are significant enough to prevent me from accepting it for publication. I am sorry to write with this disappointing news; however, I am sure that you appreciate that the conclusions of your research must be seen by the wider community to be fully supported by the data.

Having said that, should you be able to carry out all the work suggested by the referees, then I would be happy to see the paper again, as a new submission. If after considering the feedback, you instead decide to submit elsewhere, please let me know, so that we can close our file.

Reviewer 1

Comments for the author

The manuscript "Gap junction-mediated signaling coordinates Rhodopsin coupling for *Drosophila* color vision" addresses the role of genes encoding cell-junction proteins in the cell interactions that occur between photoreceptor and accessory cells to regulate rhodopsin gene expression in R7 and R8. The authors identify a requirement of *Shak-B*, *Inx2* and *Inx7* in ommatidial accessory cells to regulate rhodopsin gene expression in R8 cells. Experimentally, the authors study in pupal eyes the

ratio of ommatidia expressing normal rhodopsin combinations in R7 and R8 (Rh4/Rh6 and Rh3/Rh5) or the Rh3/Rh6 abnormal combination, after the knockdown of several genes encoding cell-junction proteins (adherence junction, septate junction and gap junction). They identify a requirement for *Inx2*, *Ink7* and *Shk-B*. Using a Gal4 driver expressed in cone cells and in PPCs, they show that this requirement is restricted to these accessory cells. Furthermore, using conditional expression of Gal80 they are able to restrict the time window when this requirement is effective. The authors also analyze the possible involvement of Notch signaling, and their results confirm that correct development of accessory cells is important to generate the correct pairing of rhodopsin expression in R7/R8. Finally, the authors study the consequences of IP3R and PKA in rhodopsin coupling, and find that loss of IP3R causes miss-coupling. This last part of the manuscript is a bit confusing, with a mix of results and discussion that is not fully consistent. I would suggest to explain first why is important to check the requirements of these genes (IP3R and PKA) in the context of gap-junction formation in the *Drosophila* eye, then describe the results, and finally describe some examples where this link is also relevant. Altogether, the main claim of the manuscript ("The Rh5-inducing instructive signal from R7 to R8 is regulated by signaling molecules that are transported from adjacent accessory cells to PR cells through gap junctions." is a bit of an overstatement, as what they really show is that gap junctions play a significant role in the Rh5-inducing instructive signal from R7 to R8. It may be that they are required to transport signaling molecules from adjacent accessory cells to PR, as the authors claim, or that they are important for the normal development of accessory cells, and consequently affect indirectly the communication between these cells and the PR. This seems to be the case of Notch.

Minor:

- Spa-Gal4 ? If the authors refer to spalt the correct abbreviation is sal.
- All UAS-RNAi lines in material and methods lack proper identification (page 9).
- Show the expression of the Gal4 line used to drive expression in accessory cells (spa-Gal4).
- They may use some other Gal4 line with expression restricted to either the cone cells or the pigmentary cells.
- Calcium waves have been recently described in the *Drosophila* retina affecting supporting cells and propagated through gap junction channels (DOI: 10.1126/science.ady5541)

Reviewer 2

Comments for the author

The manuscript by Nakagoshi and colleagues uses the developing eye primordium of *Drosophila* to unravel a role of gap-junctions in the tightly coupled expression of Rhodopsins in those photoreceptors aimed at recognizing color. They use the GAL4/UAS system to deplete expression of genes encoding for gap junction proteins *Innexin 2* and *7* in accessory cells (cells that surround and keep in touch with each ommatidia consisting of eight photoreceptors) and show that coupling in nearby photoreceptors is affected. They also present evidence that interfering with Notch and EGFR, two receptors involved in the specification of accessory cells, also phenocopies the effects of *Innexin* depletion on rhodopsin coupling in R7 and R7 photoreceptors. The paper deals with an interesting aspect of developmental signaling and the potential role of gap junctions in this process. However, the subject (Rhodopsin coupling) is not properly introduced, paper is in many aspects difficult to follow, data are highly preliminary, it is mechanistically unclear how gap junctions in accessory cells are required for Rhodopsin coupling in nearby photoreceptors, controls are lacking, and the temporally controlled depletion of genes gives rise to contradictory results. Here I will elaborate a little bit more my concerns:

- (1) The explanation of cell fate specification and rhodopsin expression in the Introduction is difficult to understand. Timing (APF) of fly development is not properly introduced and is difficult to follow for a non-fly expert.
- (2) Controls in figure 1 to show rhodopsin expression in wild type eyes are not shown. Some data are not referenced to any figure (lines 22-25, pg5)

(3) Lines 39-44, pg 6: Rhodopsin coupling: "the required time window...seems to be in the early to mid-pupal stages". Pupal stage last for 5 days and their experiments point to a requirement in the first 24 hours. This is early and not mid.

(4) Line 5, pg 7: "late third instar larval instar (18 hAPF)" does not make any sense.

(5) The temperature-controlled depletion of Notch activity gives rise to different results depending on the experiments. While gap junctions are required in the first 24 hours of pupal development for rhodopsin coupling, loss of Notch (with a Notch-DN) during this time had not effect, whereas Nts did have.

(6) Many code numbers of fly stocks are lacking and many stocks are not referenced properly

Reviewer's Responses to Questions

Experimental quality

Does each figure have the proper controls?

If 'No', please indicate reasons in Comments for Author box below.

Reviewer #1:

- Yes

Reviewer #2:

- No

Were the data analyzed using appropriate statistical tests?

If 'No', please indicate reasons in Comments for Author box below.

Reviewer #1:

- Yes

Reviewer #2:

- Yes

Reproducibility

Were experiments performed using adequate number of biological replicates?

If 'No', please indicate reasons in Comments for Author box below.

Reviewer #1:

- Yes

Reviewer #2:

- Yes

Does the methods section provide sufficient detail to permit reproducibility?

If 'No', please indicate reasons in Comments for Author box below.

Reviewer #1:

- No

Reviewer #2:

- No

Completeness

Are the manuscript's conclusions supported by the data?

If 'No', please indicate reasons in Comments for Author box below.

Reviewer #1:

- No

Reviewer #2:

- No

Scholarship

Do the authors cite and discuss the merits of data that would argue for and against their conclusion?

If 'No', please indicate reasons in Comments for Author box below.

Reviewer #1:

- Yes

Reviewer #2:

- Yes

Does the manuscript title & abstract accurately reflect the contents of the manuscript, without hyperbole?

If 'No', please indicate reasons in Comments for Author box below.

Reviewer #1:

- Yes

Reviewer #2:

- Yes

Author response to reviewers' comments

Responses to reviewers' comments.

Our responses are marked in blue. Description in the text is marked in magenta.

Reviewer 1: The manuscript "Gap junction-mediated signaling coordinates Rhodopsin coupling for *Drosophila* color vision" addresses the role of genes encoding cell-junction proteins in the cell interactions that occur between photoreceptor and accessory cells to regulate rhodopsin gene expression in R7 and R8. The authors identify a requirement of Shk-B, Inx2 and Inx7 in ommatidial accessory cells to regulate rhodopsin gene expression in R8 cells. Experimentally, the authors study in pupal eyes the ratio of ommatidia expressing normal rhodopsin combinations in R7 and R8 (Rh4/Rh6 and Rh3/Rh5) or the Rh3/Rh6 abnormal combination, after the knockdown of several genes encoding cell-junction proteins (adherence junction, septate junction and gap junction). They identify a requirement for Inx2, Inx7 and Shk-B. Using a Gal4 driver expressed in cone cells and in PPCs, they show that this requirement is restricted to these accessory cells. Furthermore, using conditional expression of Gal80 they are able to restrict the time window when this requirement is effective. The authors also analyze the possible involvement of Notch signaling, and their results confirm that correct development of accessory cells is important to generate the correct pairing of rhodopsin expression in R7/R8. Finally, the authors study the consequences of IP3R and PKA in rhodopsin coupling, and find that loss of IP3R causes miss-coupling. This last part of

the manuscript is a bit confusing, with a mix of results and discussion that is not fully consistent. I would suggest to explain first why is important to check the requirements of these genes (IP3R and PKA) in the context of gap-junction formation in the *Drosophila* eye, then describe the results, and finally describe some examples where this link is also relevant.

According to the Reviewer's comment, we reconstructed the description of the last section.

We firstly explained the reason why we focused on IP3R and PKA pathways along with several reference papers of gap junction-mediated cell differentiation as follows.

“Gap junctions can act as channels that exchange ions and small molecules directly, and calcium signaling or cAMP transportation through gap junctions regulates several types of cell differentiation. For example, gap junction-mediated calcium signaling regulates blood progenitor cell fate decision in hematopoiesis ----

To check the effect of calcium signaling and cAMP pathways on Rhodopsin coupling, a dominant-negative form of IP3 receptor (IP3R) or protein kinase A (PKA) was induced. Knocking down of IP3R, but not of PKA, apparently induced Rh3/Rh6 mis-coupling (Fig. 4A).”

Subsequently, we discussed functions of IP3R and calcium waves along with several reference papers. A recent paper suggested by the reviewer is also included.

“A recent report clearly shows that retinal calcium waves in accessory cells coordinate uniform tissue patterning through gap junctions (Choi et al., 2025). Thus, it is assumed that *Inx2*- and *Inx7*-mediated calcium signaling is involved in pigment cell development and subsequent specification of PR cells to establish Rhodopsin coupling. Although retinal calcium waves are observed in accessory cells but not in photoreceptor cells (Choi et al., 2025), an intriguing possibility for Rhodopsin coupling is that gap junction channel-associated molecules in accessory cells act as gatekeeper to control the precise timing of channel opening to PR cells and regulate subsequent PR cell maturation.”

Finally, we discussed a critical period for gap junction-mediated establishment of the instructive signal.

If the completion of accessory cell development at around 40h APF triggers transportation of signaling molecules from accessory cells to PR cells, it is consistent with the following observations. Temperature shift from 24h APF appears to be insufficient to fully knockdown *Inx2* or *Inx7* activity at around 40h APF, whereas earlier temperature shift from 0h or 12h APF substantially induced Rh3/Rh6 mis-coupling (Fig.2B, C). Therefore, it is assumed that a critical period for gap junction-mediated establishment of the instructive signal is just around the end of accessory cell development. This mechanism allows PR cells to properly start rhabdomere morphogenesis and Rhodopsin coupling after completion of accessory cell development.

Altogether, the main claim of the manuscript (“The Rh5-inducing instructive signal from R7 to R8 is regulated by signaling molecules that are transported from adjacent accessory cells to PR cells through gap junctions.” is a bit of an overstatement, as what they really show is that gap junctions play a significant role in the Rh5-inducing instructive signal from R7 to R8. It may be that they are required to transport signaling molecules from adjacent accessory cells to PR, as the authors claim, or that they are important for the normal development of accessory cells, and consequently affect indirectly the communication between these cells and the PR. This seems to be the case of Notch.

According to the Reviewer's comment, we toned down the description.

“The Rh5-inducing instructive signal from R7 to R8 is regulated by signaling molecules that are transported from adjacent accessory cells to PR cells through gap junctions.” was changed to the following description.

Our results strongly suggest that the Rh5-inducing instructive signal from R7 to R8 is regulated by adjacent accessory cells through gap junction-mediated signaling.

Furthermore, subtitle of the 3rd section “Notch-mediated accessory cell development establishes the instructive signal” was also toned down and changed to the following description. Title of Fig. 3 was also changed.

Notch-mediated accessory cell development correlates with the instructive signal

Minor:

-Spa-Gal4 ? If the authors refer to spalt the correct abbreviation is sal.

Spa-GAL4 is described in Jiao et al. (2001) Development 128, 3307-3319.

spa-Gal4 was prepared by cloning the 7.1 kb *EcoRI* genomic fragment of *D-Pax2* (Fu and Noll, 1997), extending from intron 2 into intron 4 and including the *spa* enhancer, into the *NotI* site of the pDA188.1 vector (a P element vector including the *hsp70* minimal promoter, the Gal4-coding region and the *tubulina1* trailer, prepared and provided by D. Nellen and K. Basler).
<https://pubmed.ncbi.nlm.nih.gov/11546747/>

Expression in cone cells is reported in Nagaraj and Banerjee (2007) Development 134, 825-831. So, we cited this reference and showed expression pattern of spa-GAL4 in a pupal eye at 40h APF in Fig. S2.

-All UAS-RNAi lines in material and methods lack proper identification (page 9).

We described the line number of each strain as follows.

The following GAL4/UAS lines were used: *GMR-GAL4* (Bloomington Drosophila Stock Center, BL#1104), *spa-GAL4* (BL#26656), *UAS-Stinger* (BL#84277), *UAS-IP3R^[DNJ]* (BL#602868), *UAS-PKA^[DNJ]* (BL#35550), *UAS-ogre (inx1)-IR JF02595* (BL#27283), *UAS-ogre (inx1)-IR HMS02764*, *UAS-inx2-IR-2*, *UAS-inx4-IR-1*, *UAS-inx5-IR-1*, *UAS-inx6-IR-1*, *UAS-inx6-IR-3*, *UAS-Shaking-B (inx8)-IR-3*, *UAS-DE-Cad-IR-1* (NIG), *UAS-inx3-IR v39094*, *UAS-inx7-IR v22949*, *UAS-dlg-IR v41134*, *UAS-cora-IR v9788* (VDRC), and *UAS-d.n.N (N^[DNJ])* (Go et al., 1998; Tanaka et al., 2007). For temporarily regulated inhibition of *inx* and *N*, *tub-GAL80^{ts}* (BL#7017) and *N^{ts}* (BL#2533) were used. *GMR-wIR* (gift from R. Carthew) was used to induce *white* RNAi in adult eyes (Lee and Carthew, 2003). Rh4 expression was monitored with Rh4-EGFP (BL#7456).

-Show the expression of the Gal4 line used to drive expression in accessory cells (*spa-Gal4*).

We showed expression pattern of *spa-GAL4* in a pupal eye at 40h APF in Fig. S2.

-They may use some other Gal4 line with expression restricted to either the cone cells or the pigmentary cells.

We used another GAL4 line (54C-GAL4) that is expressed in pigment cells. However, *inx* KD resulted in abnormal rhabdomere formation. So, we could not examine the Rhodopsin coupling.

-Calcium waves have been recently described in the Drosophila retina affecting supporting cells and propagated through gap junction channels (DOI: 10.1126/science.ady5541)

We cited this paper in the text as described above.

Based on reviewer’s suggestion on the PDF manuscript, we changed the description throughout the manuscript. We greatly appreciate helpful reviewer’s comments.

Reference papers of connexin mutations in human diseases were added as follows.

Mutations in human connexins can lead to diseases, supporting physiological importance of gap junction channels. For example, mutations in Connexin26 (Cx26) and Cx30 are responsible for deafness (del Castillo et al., 2002; Denoyelle et al., 1998; Grifa et al., 1999; Kelsell et al., 1997),

and mutations in Cx50 and Cx46 are responsible for inherited cataract (Mackay et al., 1999; Shiels et al., 1998).

Responses to reviewers' comments.

Our responses are marked in blue. Description in the text is marked in magenta.

Reviewer 2: The manuscript by Nakagoshi and colleagues uses the developing eye primordium of *Drosophila* to unravel a role of gap-junctions in the tightly coupled expression of Rhodopsins in those photoreceptors aimed at recognizing color. They use the GAL4/UAS system to deplete expression of genes encoding for gap junction proteins Innexin 2 and 7 in accessory cells (cells that surround and keep in touch with each ommatidia consisting of eight photoreceptors) and show that coupling in nearby photoreceptors is affected. They also present evidence that interfering with Notch and EGFR, two receptors involved in the specification of accessory cells, also phenocopies the effects of Innexin depletion on rhodopsin coupling in R7 and R7 photoreceptors. The paper deals with an interesting aspect of developmental signaling and the potential role of gap junctions in this process. However, the subject (Rhodopsin coupling) is not properly introduced, paper is in many aspects difficult to follow, data are highly preliminary, it is mechanistically unclear how gap junctions in accessory cells are required for Rhodopsin coupling in nearby photoreceptors, controls are lacking, and the temporally controlled depletion of genes gives rise to contradictory results. Here I will elaborate a little bit more my concerns:

(1) The explanation of cell fate specification and rhodopsin expression in the Introduction is difficult to understand. Timing (APF) of fly development is not properly introduced and is difficult to follow for a non-fly expert.

According to the reviewer's comment, we changed the description as follows. Underlined sentences are newly added to help understanding of readers.

The *Drosophila* compound eye is composed of approximately 800 ommatidia, and every ommatidium contains a cluster of eight PR cells surrounded by accessory cells. Accessory cells consist of four cone cells, two primary pigment cells (PPCs), and a lattice of secondary (2°) and tertiary (3°) pigment cells (Fig. 1A). PR cells have light-gathering structure (rhabdomere) and are classified into two groups based on the position of the rhabdomere. Six outer PR cells (R1-R6) have their rhabdomeres in the outer position and express Rhodopsin1 (Rh1), which is involved in object (motion) detection. Two inner PR cells (R7 and R8) have their rhabdomeres in the inner position, vertically aligned on the same axis in an ommatidium, and express Rh3-Rh6 that are involved in color vision (Fig. 1B). During retinal development, specification of cone-cell fate is induced by combinatorial EGFR and Notch (N) pathways originating from PR cells (Nagaraj and Banerjee, 2007), and these cone cells have direct contact with PR cells during the four-cone cell stage and subsequent developmental stages (Fig. 1A). Thereafter, at 18h after puparium formation (APF), the two interommatidial cells adjacent to the anterior-posterior cone cells receive high N signals, resulting the expression of the immunoglobulin cell adhesion molecules Hibris (Hbs) and Sticks-and-Stones (Sns) to form PPCs (1°, light brown in Fig. 1A) (Bao, 2014). Interommatidial lattice of 2° and 3° pigment cells are formed with N-mediated signals to remove unneeded cells, while cone cells and PPCs oppose this signal through activation of EGFR signaling (Miller and Cagan, 1998). At 40h APF, differentiation of all accessory cells is completed (Fig. 1A). Subsequently, rhabdomere morphogenesis and expression of rhodopsin genes are induced during the late pupal stage (Earl and Britt, 2006). R7 expresses stochastically either of the UV-sensitive Rh3 or Rh4, whereas R8 expresses either of blue-sensitive Rh5 or green-sensitive Rh6. Rh3 expression in R7 is coupled with Rh5 in R8, whereas Rh4 expression in R7 is coupled with Rh6 in R8. This Rhodopsin coupling is highly coordinated through instructive signals from R7 to R8 (Chou et al., 1999) and generates two major ommatidial subtypes, pale (Rh3/Rh5) and yellow (Rh4/Rh6) types. These two ommatidial subtypes are distributed randomly throughout the whole eye and the ratio is about 30% of pale type and 70% of yellow type (Mollereau and Domingos, 2005; Wernet and Desplan, 2004)(Fig. 1B). Activity of the homeodomain protein Defective proventriculus (Dve) is required to suppress *rh3* in yellow-type R7, and that of R1 is required to transmit the instructive signal from R7 to R8 (Johnston et al., 2011; Kitamata et al., 2024)(Fig. 1B).

Developmental time course during accessory cell development is based on description of several reports. Some examples are shown below.

Hexagonal patterning of the *Drosophila* eye
Johnson, R.I. (2021) *Dev. Biol.* 478, 173-182
(24h and 40h APF in Fig. 1A)

Notch Controls Cell Adhesion in the *Drosophila* Eye
Bao, S. (2014) *PLoS Genet.* 10, e1004087
(18h and 40h APF in Fig. 1A)

(2) Controls in figure 1 to show rhodopsin expression in wild type eyes are not shown. Some data are not referenced to any figure (lines 22-25, pg5)

Our description was insufficient to explain the control flies. To achieve immunostaining of adult eyes, fluorescence of eye pigment disturbs clear image acquisition. So, we used *GMR-white-IR* to reduce eye pigment of [*w*⁺] transgenic flies.

The ratio of mis-coupling depends on genetic backgrounds. To obtain control flies close to the test flies' genetic background, we used heterozygous flies for GAL4 and UAS lines as a control (GAL4/+ or UAS/+). Oregon-R (OR) or *w*; *GMR-wIR Rh4-EGFP* was used to obtain heterozygous control flies for GAL4 and UAS lines. We changed the description as follows and the sentence was placed after the description of Rh4-EGFP.

Oregon-R (OR) or *w*; *GMR-wIR Rh4-EGFP* was used to obtain heterozygous control flies for GAL4 and UAS lines.

We added reference papers as follows.

Cone cells form the corneal lens and cap the distal region of the rhabdomere, and they send interreticular fibers that contact the cell body of PR cells (Charlton-Perkins et al., 2021). The interaction between cone and PR cells also occurs in the four-cone cell stage during early pupal development (Tomlinson, 1985) (Fig. 1A).

(3) Lines 39-44, pg 6: Rhodopsin coupling: "the required time window...seems to be in the early to mid-pupal stages". Pupal stage last for 5 days and their experiments point to a requirement in the first 24 hours. This is early and not mid.

We changed the description as follows. Other sentences were also corrected.

Considering the time lag of recovery from GAL80-mediated blockade at 18 °C and of RNAi induction after temperature shift to 30 °C, the required time window of *Inx2* or *Inx7* activity seems to be in the early pupal stages.

(4) Line 5, pg 7: "late third instar larval instar (18 hAPF)" does not make any sense.

To adjust X-axis unit to APF (0h is a point of puparium formation) throughout the manuscript, we used minus 18h (18h before puparium formation, -18h APF). This avoids confusion using both BPF and APF. The description was changed as follows.

In contrast, temperature shift from late third larval instar (18h before puparium formation, -18h APF) showed abnormal accessory cell development associated with Rhodopsin mis-coupling (Fig. S3B, C).

(5) The temperature-controlled depletion of Notch activity gives rise to different results depending on the experiments. While gap junctions are required in the first 24 hours of pupal development for rhodopsin coupling, loss of Notch (with a Notch-DN) during this time had not effect, whereas Nts did have.

When we used *tub-GAL80^{ts}* system, temperature-shift period does not coincide with Notch inhibition. There is a time lag to fully inhibit N activity. After temperature shift, (1) inhibition of GAL80, (2) GAL4 activation, (3) GAL4-mediated transcription, (4) translation of N^[DN] protein, (5) Accumulation of N^[DN] protein to fully inactivate N signaling, should occur. This time lag is highly variable depending on levels of GAL4 expression, UAS-mediated expression, and so on. So, this kind of experiments can only determine rough time window. To help understanding of readers, we mentioned the reason for this time lag again. And we also changed the description to clearly mention the reason to use a temperature-sensitive allele Nts as "To determine the precise time window of N signal requirement for Rhodopsin coupling".

As described above, considering the time lag of recovery from GAL80-mediated blockade at 18 °C and of N^[DN] induction after temperature shift to 30 °C, inhibition of N signaling in the early pupal stage appears to affect both accessory cell development and establishment of the instructive signal. These results suggest that N-mediated accessory cell development is required to establish the instructive signal between PR cells, R7 and R8. To determine the precise time window of N signal requirement for Rhodopsin coupling, a temperature-sensitive allele of N (*N^{ts}*) was used to inhibit accessory cell development.

(6) Many code numbers of fly stocks are lacking and many stocks are not referenced properly

We described the line number of each strain as follows.

The following GAL4/UAS lines were used: *GMR-GAL4* (Bloomington Drosophila Stock Center, BL#1104), *spa-GAL4* (BL#26656), *UAS-Stinger* (BL#84277), *UAS-IP3R^[DN]* (BL#602868), *UAS-PKA^[DN]* (BL#35550), *UAS-ogre (inx1)-IR JF02595* (BL#27283), *UAS-ogre (inx1)-IR HMS02764*, *UAS-inx2-IR-2*, *UAS-inx4-IR-1*, *UAS-inx5-IR-1*, *UAS-inx6-IR-1*, *UAS-inx6-IR-3*, *UAS-Shaking-B (inx8)-IR-3*, *UAS-DE-Cad-IR-1* (NIG), *UAS-inx3-IR v39094*, *UAS-inx7-IR v22949*, *UAS-dlg-IR v41134*, *UAS-cora-IR v9788* (VDRC), and *UAS-d.n.N (N^[DN])* (Go et al., 1998; Tanaka et al., 2007). For temporarily regulated inhibition of *inx* and *N*, *tub-GAL80^{ts}* (BL#7017) and *N^{ts}* (BL#2533) were used. *GMR-wIR* (gift from R. Carthew) was used to induce *white* RNAi in adult eyes (Lee and Carthew, 2003). Rh4 expression was monitored with Rh4-EGFP (BL#7456).

ResubmissionFirst decision letter

MS ID#: bio.062463

MS Title: Gap junction-mediated signaling coordinates Rhodopsin coupling for Drosophila color vision

Authors: Xuanshuo Zhang; Ryoki Shinjo; Manabu Kitamata; Shinichi Otsune; Hideki Nakagoshi

I have now reached a decision on the above manuscript.

The reviewer reports are shown at the bottom of this email.

As you will see, the reviewers gave favourable reports, but raised some critical points that will require amendments to your manuscript. I hope that you will be able to carry these out, because we would like to be able to accept your paper.

At this stage, we also ask you to ensure your manuscript complies with our formatting guidelines - please see our manuscript preparation guidelines for details. Provided you are able to fully address the referees' comments, we are positive about publication of your paper (we accept over 95% of revision submissions) and therefore hope you won't mind any extra work involved in reformatting your manuscript at this point.

Please upload both a 'clean' version of your Word file, along with a highlighted version clearly showing where you have made changes in the revised manuscript. Please avoid using 'Track changes' in Word files as these are lost in PDF conversion.

I should be grateful if you would also provide a point-by-point response detailing how you have dealt with the points raised by the reviewers in the 'Response to Reviewers' box. Please attend to all of the reviewers' comments. If you do not agree with any of their criticisms or suggestions please explain clearly why this is so.

Reviewer 1

Comments for the author

The manuscript by Nakagoshi and colleagues uses the developing eye primordium of Drosophila to unravel a role of gap-junctions in the tightly coupled expression of Rhodopsins in those photoreceptors aimed at recognizing color. They use the GAL4/UAS system to deplete expression of genes encoding for gap junction proteins Innexin 2 and 7 in accessory cells (cells that surround and keep in touch with each ommatidia consisting of eight photoreceptors) and show that coupling in nearby photoreceptors is affected. They also present evidence that interfering with Notch, a receptor involved in the specification of accessory cells, also phenocopies the effects of Innexin depletion on rhodopsin coupling in R7 and R8 photoreceptors. Authors also present evidence of a role of Calcium in this coupling. The paper deals with an interesting aspect of developmental signaling and the potential role of gap junctions in this process. Authors have addressed my major concerns with changes in the ms. Now the paper is much easier to follow, since the Introduction has been improved, the timing of the experiments properly described and some experiments clarified.

Some issues should be addressed:

(1) When interfering with gap junction formation with the Gal80ts system, authors now use the general GMR-Gal4 driver instead of the more restricted Gal4 driver to accessory cells (eg. spa-gal4). Author should give an explanation to this change of Gal4 driver.

(2) Choi et al 2025 conclude in their abstract: "A cell type-specific "innexin code" coordinates wave propagation through a defined gap junction network among non-neuronal retinal cells, excluding photoreceptors". Shouldn't authors then change the model of Figure 4 and remove gap junctions between accessory and PR cells?, and change the description of their model in the last part of the ms? Do authors have any experimental evidence of gap junction requirement specifically in PRs? If so, please mention it.

Reviewer 2

Comments for the author

The authors have replied to most concerns raised by this reviewer, and in my view the manuscript is greatly improved and almost ready for acceptance. I have marked "NO" to the questions "Completeness" (Are the manuscript's conclusions supported by the data?) and "Scholarship" (Do the authors cite and discuss the merits of data that would argue for and against their conclusion?) because I still don't understand why the authors favor exclusively the interpretation that "the Rh5-inducing instructive signal from R7 to R8 is regulated by adjacent accessory cells through gap junction-mediated signaling". I believe that an alternative interpretation (gap junction-mediated signaling is important for the normal development of accessory cells and indirectly affect their signaling to PR cells) is still compatible with their results, and consequently they should state both alternatives.

Reviewer's Responses to Questions

Experimental quality

Does each figure have the proper controls?

If 'No', please indicate reasons in Comments for Author box below.

Reviewer #1:

- Yes

Reviewer #2:

- Yes

Were the data analyzed using appropriate statistical tests?

If 'No', please indicate reasons in Comments for Author box below.

Reviewer #1:

- Yes

Reviewer #2:

- Yes

Reproducibility

Were experiments performed using adequate number of biological replicates?

If 'No', please indicate reasons in Comments for Author box below.

Reviewer #1:

- Yes

Reviewer #2:

- Yes

Does the methods section provide sufficient detail to permit reproducibility?

If 'No', please indicate reasons in Comments for Author box below.

Reviewer #1:

- Yes

Reviewer #2:

- Yes

Completeness

Are the manuscript's conclusions supported by the data?

If 'No', please indicate reasons in Comments for Author box below.

Reviewer #1:

- Yes

Reviewer #2:

- No

Scholarship

Do the authors cite and discuss the merits of data that would argue for and against their conclusion?

If 'No', please indicate reasons in Comments for Author box below.

Reviewer #1:

- Yes

Reviewer #2:

- No

Does the manuscript title & abstract accurately reflect the contents of the manuscript, without hyperbole?

If 'No', please indicate reasons in Comments for Author box below.

Reviewer #1:

- Yes

Reviewer #2:

- Yes

First revision

Author response to reviewers' comments

Responses to reviewers' comments.

Our responses are marked in blue. Description in the text is marked in magenta.

Reviewer 1: The manuscript by Nakagoshi and colleagues uses the developing eye primordium of *Drosophila* to unravel a role of gap-junctions in the tightly coupled expression of Rhodopsins in those photoreceptors aimed at recognizing color. They use the GAL4/UAS system to deplete expression of genes encoding for gap junction proteins Innexin 2 and 7 in accessory cells (cells that surround and keep in touch with each ommatidia consisting of eight photoreceptors) and show that coupling in nearby photoreceptors is affected. They also present evidence that interfering with Notch, a receptor involved in the specification of accessory cells, also phenocopies the effects of Innexin depletion on rhodopsin coupling in R7 and R8 photoreceptors. Authors also present evidence

of a role of Calcium in this coupling. The paper deals with an interesting aspect of developmental signaling and the potential role of gap junctions in this process. Authors have addressed my major concerns with changes in the ms. Now the paper is much easier to follow, since the Introduction has been improved, the timing of the experiments properly described and some experiments clarified.

Some issues should be addressed:

(1) When interfering with gap junction formation with the Gal80ts system, authors now use the general GMR-Gal4 driver instead of the more restricted Gal4 driver to accessory cells (eg. spa-gal4). Author should give an explanation to this change of Gal4 driver.

Knocking down of *inx2* or *inx7* with GMR-GAL4 (Fig. S1) or spa-GAL4 (Fig. 1) showed mis-coupling in similar extent (about 20%). To determine the period that *Inx* activity is required for Rh5-inducing instructive signal, we should consider time lag of the GAL80ts system (as described in the text).

We thought that GMR-GAL4 driver is more favorable than spa-GAL4. Because spa-GAL4 driver has later onset of GAL4 expression, this may cause delayed GAL4 activation (relief of GAL80 blockade) for sufficient level of GAL4-mediated UAS-*inx2*-IR or UAS-*inx7*-IR expression during the required period.

We did not include this explanation in the text to avoid confusing of readers.

(2) Choi et al 2025 conclude in their abstract: "A cell type-specific "innexin code" coordinates wave propagation through a defined gap junction network among non-neuronal retinal cells, excluding photoreceptors". Shouldn't authors then change the model of Figure 4 and remove gap junctions between accessory and PR cells?, and change the description of their model in the last part of the ms? Do authors have any experimental evidence of gap junction requirement specifically in PRs? If so, please mention it.

Because we have no experimental evidence for *Inx2/7*-ShakB channels, overall description was toned down. We changed layout of Fig. 4B and described heterotypic channels as a possible mechanism in the text (and figure legend).

< Calcium signaling is required for the instructive signal >

As retinal calcium waves are observed in accessory cells but not in photoreceptor cells (Choi et al., 2025), there are two possible mechanisms for gap junction-mediated establishment of Rh5-inducing instructive signal. (1) *Inx2*- and *Inx7*-mediated calcium signaling is required for accessory cell development, and completion of accessory cell development indirectly affect PR cell maturation independent of gap junction channel activities (Fig. 4B-(1)). (2) Homotypic junctions of ShakB between PR cells and heterotypic junctions of ShakB and *Inx2/Inx7* might mediate transportation of some permeable molecules to PR cells and trigger the instructive signal (Fig. 4B-(2)).

<Fig 4B legend>

In accessory cells, gap junction-mediated calcium waves (green circles) are induced (Choi et al., 2025). This intercommunication through gap junctions, including *Inx2/Inx7* (yellow), might regulate

accessory cell development and indirectly affect Rh5-inducing instructive signal shown as a black arrow (1). Alternatively, homotypic junctions of ShakB (blue) between PR cells and heterotypic junctions of ShakB and Inx2/Inx7 might mediate transportation of some permeable molecules to regulate the instructive signal (2).

Reviewer 2: The authors have replied to most concerns raised by this reviewer, and in my view the manuscript is greatly improved and almost ready for acceptance. I have marked "NO" to the questions "Completeness" (Are the manuscript's conclusions supported by the data?) and "Scholarship" (Do the authors cite and discuss the merits of data that would argue for and against their conclusion?) because I still don't understand why the authors favor exclusively the interpretation that "the Rh5-inducing instructive signal from R7 to R8 is regulated by adjacent accessory cells through gap junction-mediated signaling". I believe that an alternative interpretation (gap junction-mediated signaling is important for the normal development of accessory cells and indirectly affect their signaling to PR cells) is still compatible with their results, and consequently they should state both alternatives.

According to reviewer's comment, we changed layout of Fig. 4B and described two possible mechanisms in the text (and figure legend).

< Calcium signaling is required for the instructive signal >

As retinal calcium waves are observed in accessory cells but not in photoreceptor cells (Choi et al., 2025), there are two possible mechanisms for gap junction-mediated establishment of Rh5-inducing instructive signal. (1) Inx2- and Inx7-mediated calcium signaling is required for accessory cell development, and completion of accessory cell development indirectly affect PR cell maturation independent of gap junction channel activities (Fig. 4B-(1)). (2) Homotypic junctions of ShakB between PR cells and heterotypic junctions of ShakB and Inx2/Inx7 might mediate transportation of some permeable molecules to PR cells and trigger the instructive signal (Fig. 4B-(2)).

<Fig 4B legend>

In accessory cells, gap junction-mediated calcium waves (green circles) are induced (Choi et al., 2025). This intercommunication through gap junctions, including Inx2/Inx7 (yellow), might regulate accessory cell development and indirectly affect Rh5-inducing instructive signal shown as a black arrow (1). Alternatively, homotypic junctions of ShakB (blue) between PR cells and heterotypic junctions of ShakB and Inx2/Inx7 might mediate transportation of some permeable molecules to regulate the instructive signal (2).

Second decision letter

MS ID#: bio.062463R1

MS Title: Gap junction-mediated signaling coordinates Rhodopsin coupling for Drosophila color vision

Authors: Xuanshuo Zhang; Ryoki Shinjo; Manabu Kitamata; Shinichi Otsune; Hideki Nakagoshi

I am happy to tell you that your manuscript has been accepted for publication in Biology Open, pending our standard publication integrity checks. It was accepted on 12th January 2026.